# Construction of Composite Microorganisms and Their Physiological Mechanisms of Postharvest Disease Control in Red Grapes

**DOI:** 10.3390/foods14030408

**Published:** 2025-01-26

**Authors:** Jingwei Chen, Kaili Wang, Esa Abiso Godana, Dhanasekaran Solairaj, Qiya Yang, Hongyin Zhang

**Affiliations:** School of Food and Biological Engineering, Jiangsu University, Zhenjiang 212013, China; 18896670893@163.com (J.C.); 15981843773@163.com (K.W.); esa.abiso@hotmail.com (E.A.G.); solaibt@hotmail.com (D.S.); yangqiya1118@163.com (Q.Y.)

**Keywords:** red grapes, composite microorganism, biocontrol, black mold, blue mold

## Abstract

Red grapes often suffer from postharvest diseases like blue mold and black mold caused by *Penicillium expansum* and *Aspergillus niger*. Biological control using beneficial yeasts and bacteria is an effective method to manage these diseases. *Rhodotorula* sp. and *Bacillus* sp. are effective microorganisms for the control of postharvest diseases of red grapes. This study combined two yeast strains (*Rhodotorula graminis* and *Rhodotorula babjevae*) and two bacterial strains (*Bacillus licheniformis* and *Bacillus velezensis*) to investigate their biological control effects on major postharvest diseases of red grapes and explore the underlying physiological mechanisms. Research showed that compound microorganism W3 outperformed the others; it reduced spore germination and germ tube growth of *P. expansum* and *A. niger*, while its volatiles further inhibited pathogen growth. Additionally, the treatment enhanced the antioxidant capacity of grapes and increased resistance to pathogens by boosting peroxidase activities, superoxide dismutase, catalase and ascorbate peroxidase, phenylalanine ammonolyase, and polyphenol oxidase. Furthermore, the combined treatment increased the activity and accumulation of antifungal compounds such as total phenols and flavonoids, thereby improving disease resistance and reducing decay. Therefore, composite microorganisms combining various antagonistic strains may offer a viable substitute for tackling postharvest diseases in red grapes.

## 1. Introduction

Red grapes are highly valued in the commercial market due to their esthetic appeal and palatable flavor, as well as their nutritional content, which includes a rich array of vitamins, minerals, and diverse natural phytochemicals. China has been the world’s largest producer of grapes for many years [1]. Nonetheless, red grapes are highly susceptible to diseases and infections during the postharvest period, from harvest to sale, often resulting in spoilage, reduced quality, and economic losses for the industry.

Currently, techniques such as cryopreservation, regulated airflow, and irradiation preservation are commonly used physical preservation methods, with cryopreservation being the most widely applied. However, since most pathogenic bacteria can retain their virulence at low temperatures, cryopreservation alone is insufficient to inhibit their growth [2]. The application of chemicals is also a traditional means of controlling postharvest diseases in grapes. Fungicides such as natamycin and dicarboxamide [3] significantly reduce disease incidence but are susceptible to pathogen resistance and may pose risks to the environment and human health. Over the past decade, biological control technologies were extensively studied in anticipation of replacing chemical agents [4]. Therefore, it is essential to select safe and effective antagonistic microorganisms. Numerous safe and non-toxic antagonistic yeasts and bacteria show great potential in controlling postharvest diseases of red grapes [5,6].

In addition, fungi, such as *Rhodotorula* sp., are commonly employed as biocontrol agents to suppress pathogens [7,8,9,10]. *Rhodotorula* sp. is highly resilient and widely distributed in nature, known for its ability to synthesize a variety of valuable biomolecules [8]. Several studies have shown that *Rhodotorula* sp. is able to promote the growth of probiotics without affecting their own growth profile [9]. The study of Mathew et al. reported that *Rhodotorula* sp. can achieve the inhibition of pathogenic bacteria by producing torularhodin [10]. The bacterium *Bacillus* sp. is widely used as an environmentally friendly biocontrol agent, effective against a wide range of pathogens [11,12]. It was reported to produce various metabolites and antimicrobial compounds, as well as to promote plant growth [13]. Moreover, they can generate a substantial quantity of volatile organic compounds (VOCs), including alcohols, olefins, and benzenes, which exhibit potent inhibitory effects against plant pathogens [14]. For example, different VOCs produced by *B. subtilis*, such as nonane-2-one, beta-phenylethylamine, and 2-methyl-1,4-diazine, effectively control anthracnose pathogens on postharvest mangoes [15]. In conclusion, antagonistic yeasts and bacteria are capable of inhibiting pathogenic fungi through competition for nutrients and ecological niches, the secretion of various antibacterial substances, and the induction of host resistance. Notably, *Bacillus* sp. and *Rhodotorula* sp., which are extensively distributed on the surfaces of grapes, were demonstrated to enhance fruit growth and quality, representing promising candidates for the control of postharvest diseases in red grapes [16,17,18].

Yeasts and bacteria are the primary taxa utilized for the control of pathogens. In recent years, as biological control technology has advanced, the screening and utilization of antagonistic fungi and bacteria have become increasingly significant in controlling fruit and vegetable pathogens. The development of composite bacterial agents, comprising strains with diverse functions, has emerged as a prominent trend in microbiological research for postharvest disease control in fruits and vegetables [19]. Studies have shown that composite biocontrol agents, including bacterial consortia and combinations of bacteria with fungi, effectively manage plant diseases and promote plant growth. These biocontrol agents, particularly when formulated as composite inoculants, have proven instrumental in preventing postharvest diseases in fruits and vegetables. But to date, no composite strains have been reported for the prevention of postharvest blue mold and black mold in red grapes. Compared to single antagonistic strains, composite strains may exhibit complementary functions, thereby better adapting to the environment and performing more effectively [20,21]. For example, Srivastava et al. showed that *Trichoderma harzianum* and *Pseudomonas fluorescens* together were more effective in controlling tomato wilt compared to their individual use [22]. A study involving five biocontrol strains arranged in various combinations revealed that the combined strains were more effective than a single strain in controlling banana wilt [23].

Previous studies have demonstrated that composite strains exhibit superior efficacy in controlling postharvest diseases in fruits and vegetables compared to single strains. Moreover, greater variation among strains within the combination tends to enhance the biocontrol’s effectiveness. Therefore, research on the combined benefits and mechanisms of complex strains demonstrates significant potential, and these findings provide a theoretical framework and experimental evidence for the formulation of microbial fungicides. In this study, we constructed a complex of microorganisms to enhance the inhibitory effect on postharvest diseases in red grapes and explored the potential physiological mechanisms underlying their efficacy.

## 2. Materials and Methods

### 2.1. Grapes

Fresh, healthy, and uniformly ripe red grapes were purchased from the ‘Guantangqiao’ vegetable market in Zhenjiang City, Jiangsu Province, China, for subsequent experiments. The grapes were soaked in a 0.1% NaOCl solution for 3–5 min, then rinsed with tap water and placed in clean plastic baskets to dry. A hole, 3 mm in diameter, was punched in the waist of each grape using an inactivated punch, and 10 μL of a 1 × 10^8^ CFU/mL antagonistic suspension was injected into each hole. The grapes were girdled with an inactivated perforator, and 10 μL of a 1 × 10^8^ CFU/mL antagonistic yeast and bacterial suspension was injected into each hole, which was then left to dry for 2 h. Subsequently, 10 μL of a 1 × 10^5^ CFU/mL spore suspension was inoculated. After drying, the fruits were packed in plastic baskets and wrapped with plastic wrap for storage at 25 °C and 95% humidity. The decay rate and decay diameter of the fruits were counted after 3 days of storage. A total of 3 replicates were conducted, with 24 fruits in each group.

### 2.2. Pathogens

*P. expansum* and *A. niger* were isolated from rotted red grapes by our research team and preserved in our laboratory. These pathogens were cultured at 25 °C for 7 days in potato dextrose agar medium (PDA). Spores were collected with sterile physiological saline. The activated pathogens were made into a 1 × 10^5^ CFU/mL spore suspension, employing a hemocytometer and microscope, for subsequent experiments following the study of Zhu et al. [24].

### 2.3. Antagonistic Microbes

In the previous work, strains Y1 (*Rhodotorula graminis*), Y2 (*Rhodotorula babjevae*), B2 (*Bacillus licheniformis*), and B4 (*Bacillus velezensis*) were screened and found effective in preventing and controlling postharvest diseases in red grapes. The strains were deposited in the China Typical Culture Preservation Center (CCTCC) of Wuhan University with the deposit numbers Y1 (CCTCC AY 2024002), Y2 (CCTCC AY 2024003), B2 (CCTCC AB 2024149), and B4 (CCTCC AB 2024148). Therefore, these four strains were selected as antagonistic strains for the composite. Purified bacteria were inoculated in Luria–Bertani medium (LBA; containing 10 g/L tryptone, 5 g/L yeast extract, and 10 g/L NaCl) and incubated at 30 °C, 180 rpm for 16 h. Purified yeast was inoculated in nutrient yeast dextrose base (NYDB; containing 8 g/L beef extract powder, 5 g/L yeast extract, and 10 g/L glucose) and incubated at 30 °C for 24 h. The culture medium was centrifuged at 20 °C, 8000 rpm for 10 min. After removing the supernatant, the bacterial sludge was washed three times with sterile saline and prepared into a 1 × 10^8^ CFU/mL suspension using a hemocytometer.

### 2.4. Construction of Composite Strains

#### 2.4.1. Affinity Test Between Strains

After isolation and purification, the antagonistic strains were used for cross-stripping to test whether antagonism exists between different strains. Specific methods are as follows: the activated Y1, Y2, B2, and B4 strains were two by two cross-hatched on the PDA medium plate. After 3 days of incubation at 28 °C, the cross-points were observed. If the two antagonistic colonies were intermittently separated at the intersection, it indicated that the two strains were antagonistic to each other and could not be co-cultured. If the colonies at the cross-points overlapped and the two strains grew well, it indicated that the two strains had good affinity with each other and could be combined to form a composite microorganism agent.

#### 2.4.2. Best Ratio of Composite Microorganism with Inhibitory Effect on Blue Mold

The yeast Y1 and bacterium B2 with the best biocontrol effect were selected and they were compounded according to the ratio of 1:1, 1:2, 1:3, 1:4, and 1:5, respectively, by volume. Then, 10 μL of each mixture was injected into the holes of the treated red grapes. Sterile water was used as control. After two hours of drying, 10 μL of *P. expansum* and *A. niger* spore suspension was introduced into each hole of the red grapes. After drying again, the grapes were packed into plastic baskets and wrapped with plastic wrap for storage at 25 °C and 95% humidity. The decay rate and decay diameter of the fruits were counted after 3 days of storage, with 3 replicates set up, each with 24 fruits.

The composite proportion with the best control effect on blue mold of red grapes was selected and named W1, and that with the best control effect on postharvest black mold was named Q1. Then, the optimal composite proportions between the subsequent antagonistic microorganisms and composite microorganisms W1 and Q1 were explored, and these were named W2 and Q2, respectively, and so on. The final composite ratios with the best inhibitory effects on *P. expansum* and *A. niger* were determined as W3 and Q3, respectively. The whole process of constructing the composite proportions of the bacteria and yeasts were carried out on red grapes, and the in vivo effect of the composite microorganism on the control of postharvest blue mold and black mold of red grapes was explored.

#### 2.4.3. Determination of Optimal Composite Ratio

From the above obtained composite microorganism ratio W3 and Q3 used to treat the red grapes, their control effect on *P. expansum* and *A. niger* were, respectively, compared. Then, the one with the best comprehensive effect as the final composite microorganism ratio was chosen.

### 2.5. Effect of Composite Microorganisms on Spore Germination Rate and Germ Tube Length of P. expansum and A. niger

Suspensions of yeast strains Y1 and Y2, the bacterial strains B2 and B4, and the composite microorganism W3, each at a density of 1 × 10^8^ cells/mL, were inoculated into 20 mL of potato dextrose base (PDB) medium, with sterile saline serving as a control. Additionally, spore suspensions of *P. expansum* and *A. niger*, at a concentration of 1 × 10^5^ spores/mL, were individually introduced into the PDB medium. All cultures were incubated at 25 °C with a shaking speed of 75 rpm. The germination rate and germ tube length of *P. expansum* and *A. niger* were monitored at 10 h and 12 h post-inoculation, respectively.

### 2.6. Control of Red Grape Diseases by Volatile Metabolites Produced by Composite Microorganisms

#### 2.6.1. Inhibitory Effect of Volatile Metabolites Produced by Composite Microorganisms on Growth of Pathogens

The inhibitory effect of volatile metabolites produced by the composite of antagonistic microbes was determined using the double-plate-pair-buckling method [25]. A hole was punched in the center of the PDA medium plate on one side, and 50 μL of *P. expansum* and *A. niger* spore suspension (concentration of 1 × 10^5^ CFU/mL) was added. On the opposite side of the PDA medium plate, 100 μL of the microorganism suspensions of Y1, Y2, B2, B4, and W3 at a concentration of 1 × 10^8^ cells/mL, with sterile saline used as a control. After drying, the two PDA plates were docked and sealed with a sealing film. The colony diameters of the molds were determined after incubation at 25 °C for 2 days. There were three parallels per treatment and the entire experiment was repeated three times.

#### 2.6.2. Effectiveness of Volatile Metabolites Produced by Composite Microorganisms Against Postharvest Blue Mold and Black Mold of Red Grapes

Furthermore, 350 mL of PDA medium was poured into a sterile sealed box and left to solidify before being injected and coated with 1 mL each of the yeast suspensions of Y1 and Y2, B2 and B4 bacterial suspensions, and the W3 composite microbial suspension, all with a concentration of 1 × 10^8^ cells/mL, respectively, and sterile saline was used as a control. The treated red grapes were inoculated with 10 µL of 1 × 10^5^ CFU/mL of *P. expansum* and *A. niger* spore suspensions, and after they were allowed to dry out, they were placed in a sterile sealed box, and incubated at 20 °C in a sealed box for 2 days before observing and recording the rate of decay and the diameter of the red grapes. Each treatment contained three replicates, and the entire experiment was repeated three times.

#### 2.6.3. Analysis of Volatile Metabolite Fractions Produced by Composite Microorganisms

This study was conducted according to the method described by Zhu et al. with slight modification [24]. In detail, 5 mL of PDA medium was added in a sterile sealed gas bottle, and after it solidified, 20 μL each of the yeast suspensions of Y1 and Y2, B2 and B4 bacterial suspensions, and the W3 composite microbial suspension, all with a concentration of 1 × 10^8^ cells/mL, were added into each bottle, respectively. Sterile water was used as the control group. After incubation at 28 °C for 1 day, the main components of the volatile metabolites were detected and analyzed using gas chromatography–mass spectrometry (GC-MS). The assay conditions were performed according to the method of Shi et al. [26]. The mass spectrometry results were automatically retrieved from the NIST (National Institute of Standards and Technology) database, and compounds with a match score of more than 80 were retained. The retention index (RI calculated) of each compound was calculated based on the retention time and the number of carbon atoms of neighboring n-alkanes, and then further confirmed with the RI reference obtained from the literature [27]. Compounds not found in both the sterile water control and the monomicrobial or volatilized in the monomicrobial but at low levels were used as the volatile metabolite fractions of the composite microorganisms. The relative amounts of each fraction were calculated.

### 2.7. Control of Red Grape Diseases by Non-Volatile Metabolites Produced by Composite Microorganisms

#### 2.7.1. Inhibitory Effect of Non-Volatile Metabolites Produced by Complex Microorganisms on Growth of *P. expansum* and *A. niger*

Y1, Y2, B2, B4, and W3 were inoculated into the PDB liquid medium at a concentration of 1% (*v*/*v*) for each respective strain. The cultures were incubated for 48 h at 28 °C. Subsequently, the cultures were centrifuged for 10 min using a high-speed cryo-centrifuge at 12,000× *g* and 4 °C, and the supernatant was obtained by filtration twice using a sterile organic filter membrane (0.22 μm). Holes were punched in the center of the PDA solid medium, and 50 μL of supernatant from Y1, Y2, B2, B4, and W3 were added to the holes and allowed to dry. After drying, 100 μL of 1 × 10^5^ CFU/mL of *P. expansum* and *A. niger* spore suspension were added to the holes, respectively. The colonies were incubated continuously for 7 days at 25 °C, and the diameter of the mold colonies was measured. There were three parallels per treatment, and the whole experiment was repeated three times.

#### 2.7.2. Effectiveness of Non-Volatile Metabolites Produced by Composite Microorganisms Against Postharvest Blue Mold and Black Mold of Red Grapes

The treatment of red grapes followed the same procedure as in Section 2.1, and the preparation of the antagonist supernatant was also as described in Section 2.7.1. Then, 10 μL of the supernatants from Y1, Y2, B2, B4, and W3 were applied to the grapes. The control condition was established with sterile water. After 2 h to dry, 10 μL of 1 × 10^5^ CFU/mL spore suspensions of *P. expansum* and *A. niger* were inoculated. The fruits were then dried again, packed in plastic baskets, wrapped in plastic, and stored at 25 °C with 95% humidity. The decay rate and diameter were measured after 3 days of storage, with 3 replicates and 24 fruits per group.

### 2.8. Effect of Complex Microorganism W3 on Resistant Substances in Red Grapes

#### 2.8.1. The Effect of Complex Microorganism W3 on the Metabolism of Active Oxygen in Red Grapes

The preparation of the antagonistic microorganism supernatant was the same as described in Section 2.3. Red grapes were divided into six groups, with 10 μL of microorganism suspensions (1 × 10^8^ cells/mL) of Y1, Y2, B2, B4, and W3 were added to each group, respectively. The control condition was established with sterile water. After resting and drying, the fruits were packed into plastic baskets and wrapped with plastic wrap and placed in storage at 25 °C, 95% humidity for 0, 1, 2, 3, 4, and 5 days. Then, the affected area of red grapes around the wound was removed, the tissue from the wound was excised, and 0.5 g of the fruit sample was extracted from the outer edge of the wound, quickly frozen with liquid nitrogen, and then placed at −80 °C for future use.

#### 2.8.2. Effect of Complex Microorganism W3 on Peroxidase (POD), Superoxide Dismutase (SOD), and Catalase (CAT) Activities of Red Grapes

To determine the content of POD, 1 mL of pH 5.5 acetic acid–sodium acetate buffer solution (0.1 mmol/L) was used to extract the crude extract from the grapes. Then, 1.5 mL of guaiacol solution (25 mmol/L) was mixed with 0.25 mL of the crude extract and 100 μL of H_2_O_2_ (0.5 mmol/L) was added to initiate the reaction. The absorbance readings at 470 nm were taken to determine the POD content, following the method described by Zhao et al. [28].

The determination of SOD and CAT contents were conducted consistently with the process of Zhou et al. [25]. Briefly, 1 mL of extract (5 mmol/L DTT and 5% PVP) was used to prepare the crude extracts of SOD and CAT. The absorbance values at 560 nm are determined. A 50% inhibition of NBT photochemical reduction at 560 nm per minute was taken as one unit of SOD enzyme activity and the result was recorded as U/g FW. The CAT enzyme activity was quantified by considering a reduction of 0.01 absorbance units at 240 nm per minute as a single unit, with results expressed in units per gram of fresh weight (U/g FW).

### 2.9. Effect of Composite Microorganism W3 on Resistance-Related Substances and Enzyme Activities of Red Grapes

#### 2.9.1. Effect of Composite Microorganism W3 on Activities of Polyphenol Oxidase (PPO), Ascorbate Peroxidase (APX), and Phenylalanine Ammonia-Lyase (PAL) in Red Grapes

The determination of PPO and APX content was based on the method of Zhou et al. with slight modification [25]. A phosphate-buffer solution of 50 mmol/L was used to extract the crude extracts of PPO and APX, and distilled water was used as a reference to adjust to zero. The extraction method was the same as described in Section 2.8.1. PPO activity was defined as the quantity of enzyme needed to raise the absorbance at 420 nm by 1 unit per minute in the reaction mixture, with the outcome expressed in units per gram fresh weight (U/g FW). For APX activity, a reduction of 0.01 absorbance units at 290 nm per minute per gram of pulp was considered as one unit, with the findings documented in the same units (U/g FW).

Determination of PAL content was conducted according to the method of Zhang et al. with slight modifications [29]. Briefly, 0.1 mmol/L boric acid–borax buffer at pH 8.8 was used to extract the crude extract of PAL from red grapes. Then, 0.5 mL of L-phenylalanine solution (20 mmol/L) was mixed with 0.5 mL of the crude extract and kept at 37 °C for 1 h. The reaction was then terminated with 0.1 mL of 6 mol/L hydrochloric acid solution, and the absorbance value at 290 nm was measured. One unit of PAL enzyme activity was defined as the amount of enzyme necessary to enhance the absorbance at 290 nm by 0.01 units per minute within the reaction system, with the findings reported as units per gram of fresh weight (U/g FW).

#### 2.9.2. Effect of Complex Microorganism W3 on Total Phenolic and Flavonoid Contents of Red Grapes

Fruit pulp (0.5 g) was mixed with 1 mL of 1% HCl–methanol solution, extracted in an ice bath, and protected from light for 20 min. During this time, it was shaken several times, and the supernatant was collected. The absorbance values of the supernatant at 280 nm and 325 nm were determined using 1% HCl–methanol solution as a blank reference. The absorbance at 280 nm was used to represent the total phenol content per gram of fresh weight, and the absorbance at 325 nm was used to represent the flavonoid content per gram of fresh weight, according to Deng et al.’s [30] method.

### 2.10. Data Analysis

Data analysis was performed using one-way analysis of variance (ANOVA) with IBM SPSS Statistics Version 20, employing a significance level threshold of *p* < 0.05. Duncan’s multiple-range test was employed to compare means between different treatments.

## 3. Results

### 3.1. Construction of Composite Microbial Strains

#### 3.1.1. Affinity Experiment Between Antagonist Strains

As depicted in Figure 1, the strains did not produce an obvious split circle when paired, indicating no significant antagonism between the four antagonistic strains. This suggests that the strains do not compete or inhibit each other, making them suitable for the next step in the composite experiment.

#### 3.1.2. Construction of Optimal Ratio of Composite Microorganisms with Inhibitory Effects on *P. expansum* of Red Grapes

As shown in Figure 2A, among the different ratios of antagonist yeast Y1 and antagonist bacteria B2 in the composite, the group with a 1:1 ratio of Y1 to B2 exhibited the lowest percentage of red grape rot, which was significantly lower than that observed in all other groups. The decay diameter results in Figure 2B further confirm that the treatment group with the 1:1 ratio of Y1 to B2 had the lowest decay diameter, resulting in the best overall biocontrol effect. Therefore, the chosen composite ratio of Y1 to B2 was 1:1, designated as W1 (*p* < 0.05).

Figure 2C shows that the incidence of strain Y2 in the composite microorganism W1 at a ratio of 1:4 demonstrated a marked reduction compared with other groups. After five days of storage, the decay rate of the treatment group with a 1:4 composite ratio was only 23.07%, compared to 100% in the sterile water control group. Moreover, as shown in Figure 2D, the decay diameter of the treatment group with the 1:4 ratio of strain Y2 and composite microorganism W1 was significantly smaller than in the other groups. Therefore, the composite ratio of Y2 to W1 at 1:4 was chosen as W2 (*p* < 0.05).

Figure 2E,F displays the inhibitory effect of treatments with different ratios of composite microorganism W2 and strain B4 on *P. expansum* in red grapes. The outcomes indicated that the group with a 1:1 composite ratio had the lowest incidence, significantly lower than all other ratios. Its decay diameter was also the smallest, demonstrating the best inhibition of blue mold. Therefore, the final composite ratio of W2 and B4 at 1:1 was selected and designated as W3.

#### 3.1.3. Construction of Optimal Ratio of Composite Microorganisms with Inhibitory Effect on *A. niger* of Red Grapes

Similarly to the effect on blue mold, antagonistic yeast Y1 and bacterium B2 were initially used for compounding, as shown in Figure 3A. The best control of *A. niger* was achieved when the ratio of Y1 to B2 was 5:1, featuring a pronounced decrease in decay rate compared to all other groups after five days of storage. At this time, the control group experienced a complete decay rate, reaching 100%, and the decay diameter was smaller in the 5:1 ratio treatment group, as shown in Figure 3B. Therefore, we selected the 5:1 ratio of Y1 to B2, designated as Q1.

Next, we compounded the composite microorganism Q1 with yeast Y2. As shown in Figure 3C, the rot rate and diameter of spoilage were lower in the 1:1 ratio treatment compared to the other groups, and at this point, the control group reached a 100% decay rate. Therefore, we chose the 1:1 composite ratio of Q1 and Y2, designated as Q2.

Figure 3E,F shows the effect of different ratios of composite microorganism Q2 to bacterium B4 on the suppression of black mold in red grapes. The results showed that the 1:1 ratio treatment had the lowest incidence and decay diameter, significantly lower than all other ratios, demonstrating the best suppression of black mold. Therefore, the 1:1 composite ratio of Q2 and B4 was chosen as the final ratio, designated as Q3.

#### 3.1.4. Determination of Final Composite Ratio

The specific composite ratio is shown in Table 1. As shown in Figure 4A,B, both the single antagonist treatment groups and the composite microorganisms W3 and Q3 treatment groups were able to significantly reduce blue mold and black mold in red grapes, with the composite microorganism W3 treatment group having the lowest rate of decay, which was markedly less than the other groups. Compared with the composite microorganism Q3 treatment group, the W3 treatment group had a lower rate of decay and diameter of decay.

In terms of control effect on *A. niger* (Figure 4C,D), both the single antagonist treatment groups and the composite microbial treatment groups showed better results compared to the control groups. Among them, the decay rate and decay diameter of W3 were slightly smaller compared to the Q3 group.

Therefore, based on the combined effects of the two composite microorganisms, W3 and Q3, on the control of *P. expansum* and *A. niger*, we selected W3 as the optimal formulation for subsequent experiments.

### 3.2. Effect of Composite Microorganisms on Spore Germination Rate and Germ Tube Length of Pathogens

The antagonists, Y1, Y2, B2, B4, and the composite microorganism W3, presented in Figure 5, all showed inhibitory effects on *P. expansum* spore germination after 10 h in PDB medium at 28 °C and 75 r/min. They also inhibited *A. niger* spore germination after 12 h.

As shown in Figure 5A, all treatments significantly reduced the spore germination rate of *P. expansum*, with the W3 treatment exhibiting the most pronounced inhibitory effect, reducing the spore germination rate to 16.73%, compared to the control group that showed a 57.07% spore germination rate (*p* < 0.05). The control group showed the germ tube length was 34.09 μm, whereas the W3 composite treatment group was 25.91 μm. Figure 5C reveals that the composite treatment group of W3 significantly reduced *A. niger*’s spore germination to 7.5% (*p* < 0.05) and germ tube length to 45.58% of the control, which was significantly decreased in relation to the control group that showed 85% germination rate and 78.42 μm germ tube length.

### 3.3. Control of Red Grapes by Volatile Metabolites Produced by Composite Microorganisms

#### 3.3.1. Curbing Action of Volatile Excretions Produced by Composite Microorganisms on Growth of Pathogens

As shown in Figure 6A,B, the volatile metabolites of the antagonistic bacterium B4 had a significant inhibitory effect on the growth of *P. expansum* on the 3rd day. The composite microorganism W3 was more effective in inhibiting *P. expansum* than B4 on PDA medium plates, and both treatment groups were still able to effectively hinder the growth of *P. expansum* on the 4th day. This confirms that the volatile metabolites of W3, which have an inhibitory effect, are mainly volatilized by the part of B4, and the introduction of the literature suggests that B4, a bacterium, is able to volatilize substances with an inhibitory effect.

Figure 6C,D demonstrates the inhibitory effects of volatile compounds from individual antagonists and the composite microorganism W3 on the growth of *A. niger*, and the results show that both the single antagonists and the composite microorganism W3 volatiles succeeded in greatly restraining the growth of *A. niger* on the 3rd and 4th days of incubation, with W3 showing the best inhibitory effect. In this case, the volatiles emitted by W3 with an inhibitory effect are jointly produced and accumulated by each antagonist.

#### 3.3.2. Efficacy of Volatile Compounds from Composite Microorganisms in Managing Postharvest Blue Mold and Black Mold on Red Grapes

According to Figure 7, the incidence rates of postharvest blue mold and black mold on red grapes showed varying effects after fumigation with different groups. When contrasted with the control group, the postharvest blue mold on red grapes was significantly suppressed by all groups except the Y2 group (*p* < 0.05). With a rot diameter of 8.16 mm for the control group, the W3 fumigation group showed a smaller diameter of 6.83 mm, providing better control over the enlargement of lesion diameters and superior to that of the single antagonists’ fumigation group.

With a 100% incidence, the control group, as shown in Figure 7C,D, exceeded the incidence rates of the other groups; the postharvest black mold on grapes was significantly suppressed by the fumigation of the four strains of single antagonist bacteria. The incidence of black mold in grapes treated by the W3 fumigation group was markedly lower than that in the single antagonist’s fumigation groups (*p* < 0.05). With a rot diameter of 7.90 mm for the control group, the W3 fumigation group showed a smaller diameter of 6.21 mm, which provided better control over the spread of rot diameters and was superior to that of the four single antagonist fumigation treatment groups.

#### 3.3.3. Analysis of Volatile Metabolite Produced by Composite Microorganisms

The major volatile metabolites identified for the composite microorganism W3 were obtained from the results presented in Table 2. Among them, phenylethyl alcohol had the highest relative content of 30.77%; butanoic acid, 2−methyl− had the second highest relative content (7.43%); and docosanoic acid, ethyl ester was the third most abundant substance (4.63%).

### 3.4. Effects of Non-Volatile Metabolites Produced by Complex Microorganisms on Postharvest Diseases of Red Grape

#### 3.4.1. Suppression of *P. expansum* and *A. niger* Growth by Non-Volatile Metabolites Generated by Composite Microorganisms

Figure 8 shows the *in vitro* inhibitory outcomes of non-volatile metabolites yielded by different treatments on *P. expansum* (Figure 8A) and *A. niger* (Figure 8B). The results have shown that the composite microorganism treatment group exhibited slightly lower inhibition than the control group, and some individual antagonist treatment groups were also able to inhibit *P. expansum* and *A. niger*. However, no substantial differences were observed between the composite microorganism group and the individual antagonist treatment groups.

#### 3.4.2. The Impact of Non-Volatile Metabolites Produced as a Result of Composite Microorganisms on the Prevention of Blue Mold and Black Mold in Red Grapes

As shown in Figure 9, the composite microorganism treatment group was able to significantly and effectively reduce the decay rate of postharvest blue mold and black mold of red grapes compared with the control group. As shown in Figure 9A, all single antagonist treatment groups, except for the non-volatile metabolite group of antagonist bacterium B2, significantly reduced the postharvest decay rate of blue mold in red grapes, with no significant differences among the composite microorganism treatment groups. Antagonist yeast Y1 was particularly effective, with a rot diameter of only 4.54 mm compared to 6.84 mm in the control group (Figure 9B). Figure 9C reveals that the non-volatile metabolites of composite microorganisms were most effective against black mold, significantly outperforming the control group but not differing significantly from groups B2 and B4. Antagonist yeast Y2 also significantly reduced the rot rate of black mold in red grapes. Figure 9D shows that antagonist treatments Y2, B2, and the non-volatile metabolites of composite microorganisms all reduced the rot diameter of black mold, lacking notable distinctions between them (*p* < 0.05).

### 3.5. Effect of Complex Microorganism W3 on POD, SOD, and CAT Activities of Red Grapes

As shown in Figure 10A, during storage, the POD content of red grapes in the composite treatment group W3 fluctuated, increasing over the first 4 days and peaking on the 5th day. It was consistently higher than other groups. The control group (CK) showed small fluctuations in the first 4 days but increased sharply on the 5th day. Among the single antagonist groups, POD content generally increased, with slight decreases in group Y1 on the 2nd day and in groups Y2, B2, and B4 on the 4th day. For most of the first 4 days, the POD content in the treatment groups was higher than the group of CK.

As shown in Figure 10B, SOD activity in all groups followed a similar trend as it decreased initially and then increased. The composite treatment group W3 had significantly higher SOD activity than the group of CK on the 1st and 5th days and was also higher than the single antagonist groups on most days, except the 4th day (*p* < 0.05).

As shown in Figure 10C, CAT activity in the composite treatment group W3 increased at first, peaking on the 3rd day, and then decreased. The control group exhibited a general pattern of rising initially and then declining, reaching its zenith on the second day. The composite treatment group W3 generally had significantly higher CAT activity than the control and single antagonist groups, except on the 1st and 5th days (*p* < 0.05). The CAT activities of the four single antagonist treatment groups were also higher than those of the control group for most of the storage time, except for groups Y2 and B2 on the 2nd day and group B2 on the 3rd day, which were marginally below the levels observed in the control group.

### 3.6. Effects of Complex Microorganism W3 on Resistance-Related Substances and Enzyme Activity of Red Grapes

#### 3.6.1. Effects of Complex Microorganism W3 on PPO, APX, and PAL Activities of Red Grapes

As shown in Figure 11A, the PPO activities of the composite microbial group W3 were all significantly higher than those of the control group and the remaining four groups of single antagonist treatment groups in the first 4 days (*p* < 0.05). Moreover, on the first day, PPO activities peaked in both the control group and the composite treatment group W3, with the latter showing a PPO activity of 10.13 U/g FW, which was 1.32 times greater than the 7.7 U/g FW observed in the control group. Overall, the PPO activities across the four single antagonist treatments exhibited a pattern of initial increase followed by a decline. Throughout most of the storage duration, these activities were surpassed by those in the composite treatment group but exceeded the levels found in the control group.

As shown in Figure 11B, the composite treatment group W3 reached the maximum value on the 1st day and was dramatically higher than the control group throughout the storage period, and the APX activity of the composite treatment group W3 was also considerably more than that of the remaining four groups of the single antagonist groups, except for the 2nd day (*p* < 0.05). The group of CK showed a tendency of increasing and then decreasing and reached the maximum value on the 2nd day. The control group was also less than the single antagonist treatment groups in the rest of the storage period, except that it was lightly greater than the Y1, Y2, and B2 groups on the 2nd day, and slightly higher than the Y1 treatment group on the 3rd day.

As shown in Figure 11C, all treatment groups and the control group presented an ascending then descending curve. Except for the PAL activity of the composite microorganism W3 treatment group, which was lightly greater than that of the single antagonist treatment group on the 1st day, the composite treatment group W3 was notably higher than the rest of the groups and the group of CK during the entire storage period (*p* < 0.05). The PAL activity of the Y1 treatment group reached its highest value on the 3rd day, and that of the Y2, B2, and B4 treatment groups reached its highest value on the 1st day.

#### 3.6.2. Effects of Complex Microorganism W3 on Total Phenol and Flavonoid Contents of Red Grapes

Figure 12 shows the effect of different treatment groups on the total phenolic and flavonoid contents of grapes. As shown in Figure 12A, as a whole, the total phenolic content of red grapes from each treatment group exhibited a pattern of initial rise followed by a subsequent decline over the course of storage. The total phenolic content of red grapes from both the control and composite microorganism treatment groups reached the maximum on the 4th day, but the total phenolic content of red grapes from both the Y1 treatment and composite microorganism treatment groups was slightly above that of the control group; in the vast majority of the time during the whole storage period, the total phenolic content of red grapes from all groups except the control group was narrowly above that of the control group, and in most cases, the total phenolic content of red grapes from the composite microorganism group was greater than that of all the total phenolic content of red grapes from the single antagonist treatment groups.

From Figure 12B, it can be found that during the whole storage process, except for the initial storage in the presence of the Y1 treatment group on the first day and the presence of the B4 treatment group on the second day, the quantity of flavonoids in the red grapes is slightly less than the group of CK. The rest of the time, the flavonoid content of the red grapes of all the treatment groups was higher than the control group. Moreover, the majority of the time, the flavonoid content of the composite microorganism W3 treatment group of the red grapes was substantially more than all the single antagonist groups.

## 4. Discussion

Postharvest diseases of fruits and vegetables caused by fungi are one of the main causes of postharvest economic losses of fruits and vegetables [31]. The common pathogens that can cause postharvest diseases of red grapes mainly include *Strychnospora*, *P. expansum*, *A. niger*, etc. Currently, emerging biocontrol strategies for the management of postharvest diseases in fruits and vegetables include the use of yeast or bacteria with antagonistic properties; however, relying on a single strain is insufficient to protect all types of produce from decay, and the activity of some bacteria declines with the increase in storage time. Therefore, the use of composite microorganisms, which combine multiple antagonistic strains, is considered an ideal approach [32]. Past studies demonstrate that the biocontrol effect of composite microorganisms is indeed better than that of single antagonists [33]. According to the study by Cheng et al., the composite microorganisms obtained by combining three different types of *Bacillus* sp. significantly reduced the occurrence of postharvest gray mold in tomatoes, particularly in the later stages of storage [34]. Similarly, the combination of *Debaryomyces hansenii* and *Stenotrophomonas rhizophila* significantly inhibited both the occurrence rate and decay diameter of *Fusarium proliferatum* in muskmelons [35]. This study conducted a composite investigation of two antagonistic yeast strains (Y1: *R. graminis*; Y2: *R. babjevae*) and two antagonistic bacterial strains (B2: *B. licheniformis*; B4: *B. velezensis*) and explored the physiological mechanisms of postharvest disease control by composite microorganisms.

In our study, two strains of antagonistic bacteria and antagonistic yeasts both exhibiting strong biocontrol effects were initially combined. The control effect of the obtained compound microorganism on *P. expansum* and *A. niger* was dramatically higher than those of these two single antagonistic strains. Then, another yeast and a bacterium were combined with them in turn. Finally, the control effect of the composite microorganism composed of four strains on *P. expansum* and *A. niger* was not only better than that of the four single antagonist strains, but also better than that of the composite microbiome of two strains and that of the composite microbiome of three strains. Therefore, the biocontrol effect of a microbiota composed of multiple functional strains shows greater potential, which is similar to the results reported by Du et al. in their use of five biocontrol bacteria to suppress banana wilt [23].

The role of antagonistic microorganisms on pathogenic fungi is often multifaceted. Fungal infection is a common form of postharvest diseases in fruits and vegetables. Spores of pathogenic fungi can enter the internal tissues of fruits through wounds or damaged parts of the fruit surface via germ tubes produced during their germination. The vitality of spores largely reflects the pathogenicity of the pathogenic fungi [36]. The results of the experiment on the inhibition rate of *P. expansum* and *A. niger* and the spore germination rate showed that the compound microorganism could inhibit the growth and propagation of pathogenic bacteria on red grapes and the spore germination and spore tube growth in PDB medium better than the single antagonist strain. Czarnecka et al. showed that *Debaryomyces hansenii* and *Wickerhamomyces anomalus* could significantly inhibit the mycelium growth of *Monilinia fructicola* [37]. In addition to directly inhibiting pathogens, antagonistic microorganisms can also suppress pathogenic fungi by producing resistance substances. It was found that most of the antagonistic yeast and antagonistic bacteria can produce a series of antibacterial substances. Francesco et al. found that volatile organic compounds produced by two strains of *Aureobasidium pullulans* could inhibit spore germination of various molds such as *B. cinerea* and *P. expansum* [38]. The volatile metabolite produced by *Bacillus siamensis* can effectively inhibit the occurrence of gray mold of blueberry [7]. According to Rivas-Garcia et al., the hydrolase produced by the mixed culture of *Debaryomyces hansenii* and *Stenotrophomonas rhizophila* can inhibit *Fusarium proliferatum* to some extent [35]. Therefore, in our study, the volatile metabolites and non-volatile metabolites of single antagonistic strains and complex microorganisms were also studied. The findings revealed that the volatile compounds produced by the two strains of single antagonistic bacteria *Bacillus* had a certain inhibitory effect on *P. expansum* and *A. niger*, and the compound microbial volatile substances had a better control effect on *P. expansum* and *A. niger* than the two strains of single antagonistic yeasts and bacteria. Therefore, we examined the volatile gases produced by them, and after removing the control, phenylethanol with the highest content was proved. This volatile compound was reported for its inhibitory effect on the mycelium produced by *Aspergillus carbonarius* [39]. Similarly to our results, Tian et al. demonstrated that plant growth-promoting microbes (PGPMs), comprising *R. graminis* JJ10.1, *Pseudomonas psychrotolerans* YY7, and *P. chlororaphis* T8, in conjunction with *B. amyloliquefaciens* FZB42, produce volatile phenazines that effectively inhibit tomato blight and promote plant growth. The volatile phenazines produced by *B. amyloliquefaciens* FZB42 are particularly effective in suppressing tomato wilt and enhancing plant growth [21]. Similarly, Koilybayeva et al. revealed that butanoic acid, specifically 2-methylbutanoic acid, which is volatilized by *B. amyloliquefaciens*, enhances the antimicrobial efficacy of the microbial strains [40]. Moreover, 2-methylbutanoic acid constituted a significant proportion of the volatiles detected in our study. We also investigated the inhibition of *P. expansum* and *A. niger* by non-volatile substances produced by various treatments. The results showed that the growth of pathogens on PDA medium and red grapes could be reduced by the partial single antagonist treatment group and compound microorganism treatment group. However, no substantial distinction was observed between the treatment with a consortium of microorganisms and the treatment with individual antagonists. Consequently, it is reasonable to speculate that one mechanism by which complex microorganisms control the primary postharvest diseases of red grapes involves the production of volatile organic compounds (VOCs) with antibacterial properties.

In addition to their ability to produce antimicrobial substances, antagonistic microorganisms often exert their biocontrol effectiveness by enhancing host resistance. Numerous studies have demonstrated that antagonistic microorganisms can induce an increase in the activity of related defense enzymes and improve the antioxidant capacity of fruits and vegetables [41]. POD, SOD, CAT, and APX are essential antioxidant enzymes that regulate the metabolism of reactive oxygen species (ROS), prevent the synthesis of ROS, and catalyze the decomposition of H_2_O_2_ into non-toxic H_2_O and O_2_ in fruit and vegetable cells. Therefore, they play an important role in the detoxification process of ROS produced when fruits and vegetables are damaged by pathogens or other external factors [25]. Moreover, POD plays a part in fortifying the plant’s cell walls and increasing its resistance to diseases through its participation in the production of lignin and phytochemicals [28]. SOD and CAT are crucial for preserving the regular cellular activities and the equilibrium of reactive oxygen species within the cell [42]. In this study, it was found that red grapes treated with composite microorganisms could improve the antioxidant capacity of red grapes and enhance the resistance of red grapes to pathogens better than those treated with single antagonistic yeast and bacteria. Huang et al. observed that the *M. guilliermondii* Y-1 treatment boosted the activities of POD, SOD, and CAT in apple tissue, strengthened the fruit’s capacity to eliminate excessive ROS, and minimized membrane lipid peroxidation [43]. Total phenols and flavonoids are secondary metabolites in fruit and vegetable tissues, which can protect the host from pathogen infection. Benbouguerra et al. found that the antioxidant capacity of grape skin extract was highly correlated with the content of phenolic substances, and the oxidation of phenolic compounds such as flavonoids, p-coumaric acid, and ferulic acid would lead to an increase in antioxidant activity [44]. The results of our study revealed a similar phenomenon. Following treatment with complex microorganisms, the activities of POD, SOD, and CAT, as well as the contents of total flavonoids, were significantly elevated. Consequently, we can reasonably infer that complex microorganisms enhance their biocontrol efficacy by augmenting the host’s defensive capabilities and antioxidant stress resistance.

PAL and PPO are the key enzymes in the metabolic pathway of phenylpropane metabolism, and the main rate-limiting enzyme is PAL, which can be used as an index of plant infection by pathogenic bacteria [45]. PPO initiates the synthesis of quinones, compounds that exhibit toxicity to pathogens, which in turn helps to fend off pathogenic infections [25]. Koilybayeva et al. also discovered that a complex microbiota composed of three strains of *B. velezensis*, *B. subtilis*, and *B. amyloliquefaciens* could enhance polyphenol oxidase (PPO) activity in tomato fruits. The efficacy of the composite microbial treatment group was superior to that of the single antagonistic microbe treatment. Similarly, our results showed that the composite microbial treatment could effectively increase the contents of total phenols, flavonoids, PAL, and PPO in red grapes, thereby improving the disease resistance and stress resistance of the fruit and reducing the rate of fruit decay. Similar studies by Xiao et al. showed that GABA-treated *S. pararoseus* Y16 can effectively improve the activities of PPO, POD, PAL, and CAT in grapes, which is consistent with our present findings [28]. Therefore, it can be inferred that the composite microorganisms significantly enhance the activity of resistance-related enzymes in red grapes, thereby improving the disease resistance of the fruit.

## 5. Conclusions

In summary, the complex microorganism composed of a variety of antagonistic yeasts and antagonistic bacteria can effectively inhibit *P. expansum* and *A. niger*. This study provided a theoretical basis for the development and utilization of complex microorganisms in the postharvest biological control of red grapes. However, the control effect of composite microorganisms on pathogenic fungi at the molecular level remains to be further studied.

## Figures and Tables

**Figure 1 foods-14-00408-f001:**
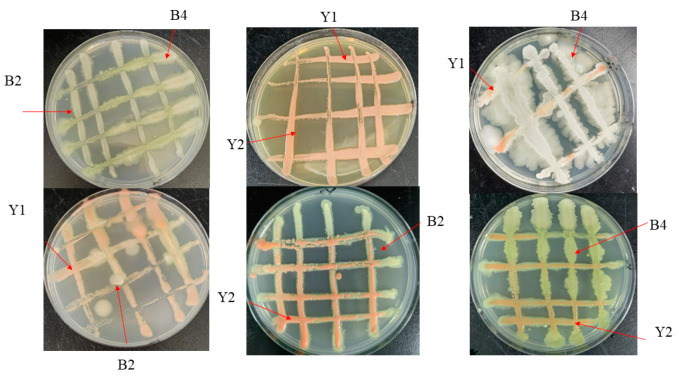
Affinity test between antagonistic strains. Note: Different letters represent different strains, Y1 (*Rhodotorula graminis*), Y2 (*Rhodotorula babjevae*), B2 (*Bacillus licheniformis*), and B4 (*Bacillus velezensis*).

**Figure 2 foods-14-00408-f002:**
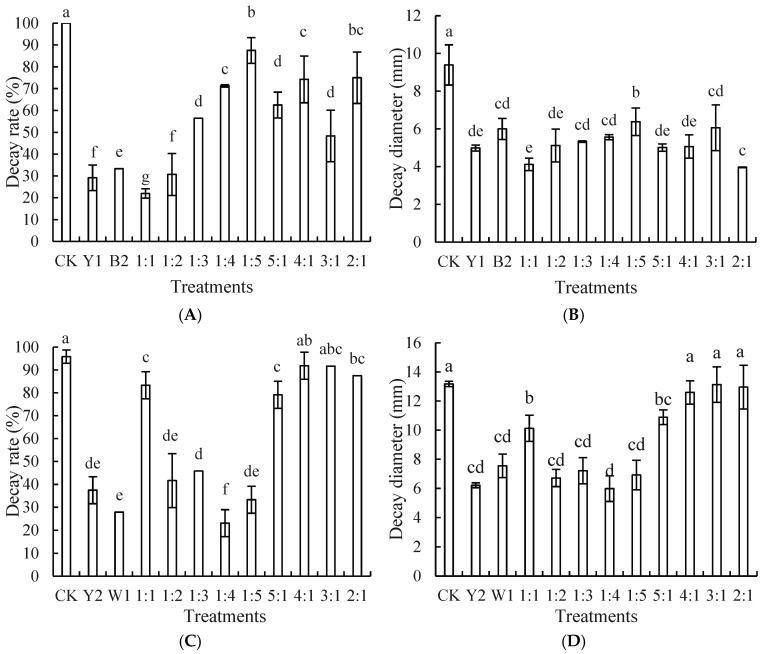
Optimal proportion construction of complex microorganisms that inhibit blue mold disease of red grapes. Note: CK is sterile water control group. (**A**,**B**) represent combination of strain Y1 with B2, (**C**,**D**) represent combination of strain W1 with Y2, and (**E**,**F**) represent combination of strain W2 with B4. Different letters represent significant differences (*p* < 0.05) according to Duncan’s multiple-range mean comparison test.

**Figure 3 foods-14-00408-f003:**
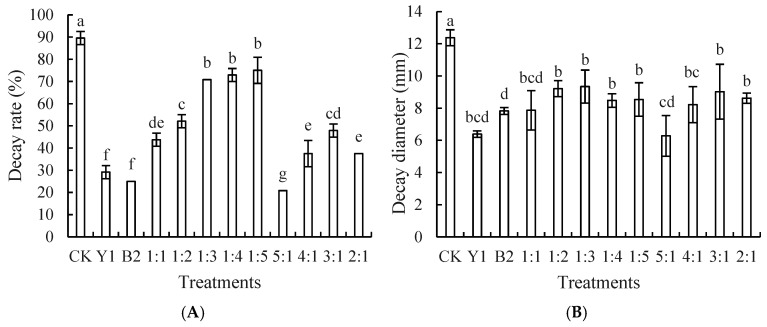
Optimal proportion construction of complex microorganisms that inhibit black mold disease of red grapes. Note: CK is sterile water control group. (**A**,**B**) represent combination of strain Y1 with B2, (**C**,**D**) represent combination of strain Q1 with Y2, (**E**,**F**) represent combination of strain Q2 with B4. Different letters represent significant differences (*p* < 0.05) according to Duncan’s multiple-range mean comparison test.

**Figure 4 foods-14-00408-f004:**
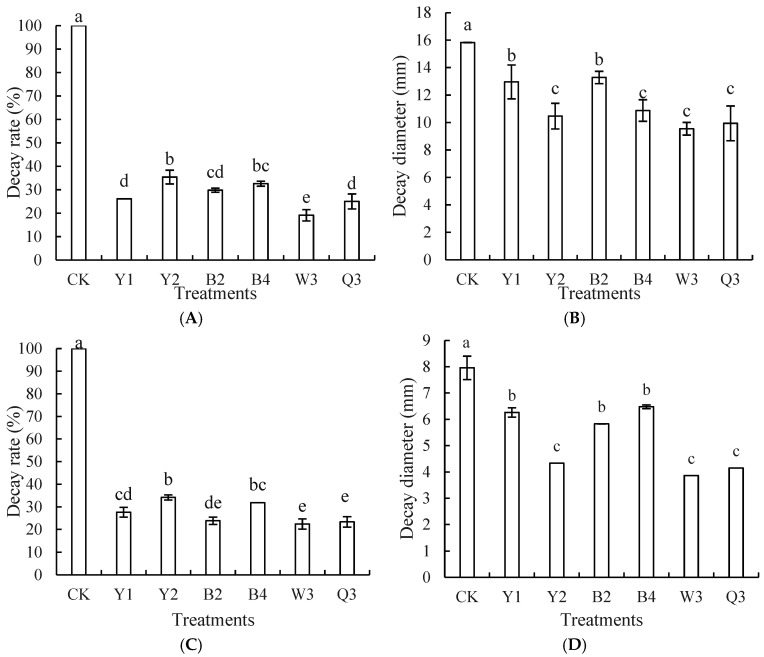
Control effects of different treatment groups blue mold (**A**,**B**) and black mold (**C**,**D**) of red grapes. Note: CK is sterile water control group; Y1, Y2, B2, B4, W3, and Q3 represent different treatment groups. Different letters represent significant differences (*p* < 0.05) according to Duncan’s multiple-range mean comparison test.

**Figure 5 foods-14-00408-f005:**
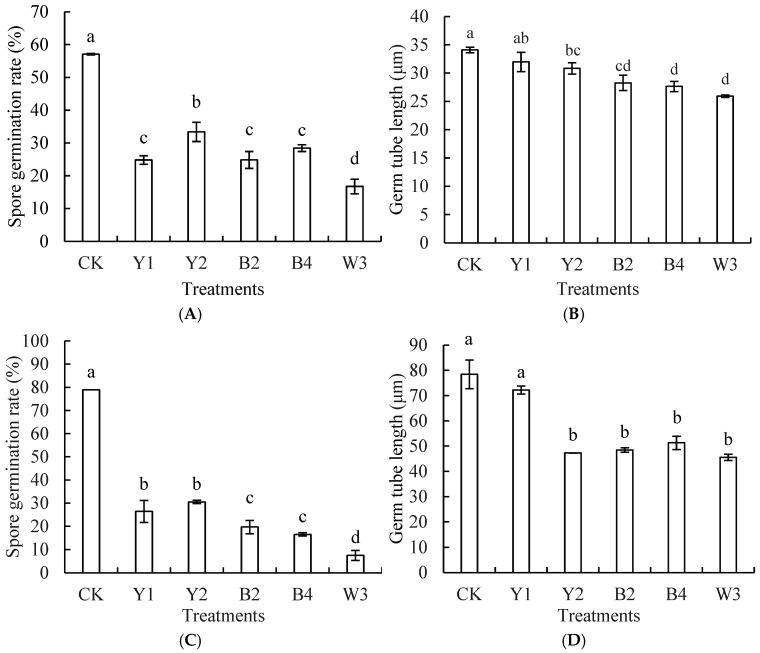
Effects of different treatments on spore germination rate and germ tube length of *P. expansum* (**A**,**B**) and *A. niger* (**C**,**D**). Note: CK is sterile water control group; Y1, Y2, B2, B4, and W3 represent different treatment groups. Different letters represent significant differences (*p* < 0.05) according to Duncan’s multiple-range mean comparison test.

**Figure 6 foods-14-00408-f006:**
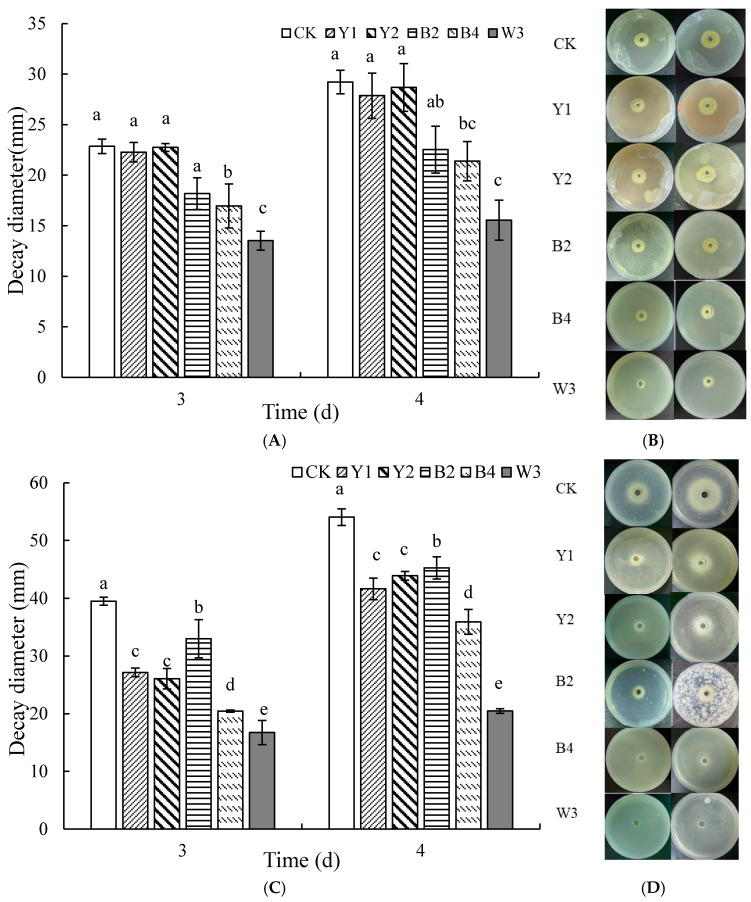
Effects of volatile metabolites produced by different treatments on *P. expansum* (**A**,**B**) and *A. niger* (**C**,**D**). Note: CK is sterile water control group; Y1, Y2, B2, B4, and W3 represent different treatment groups. Different letters represent significant differences (*p* < 0.05) according to Duncan’s multiple-range mean comparison test.

**Figure 7 foods-14-00408-f007:**
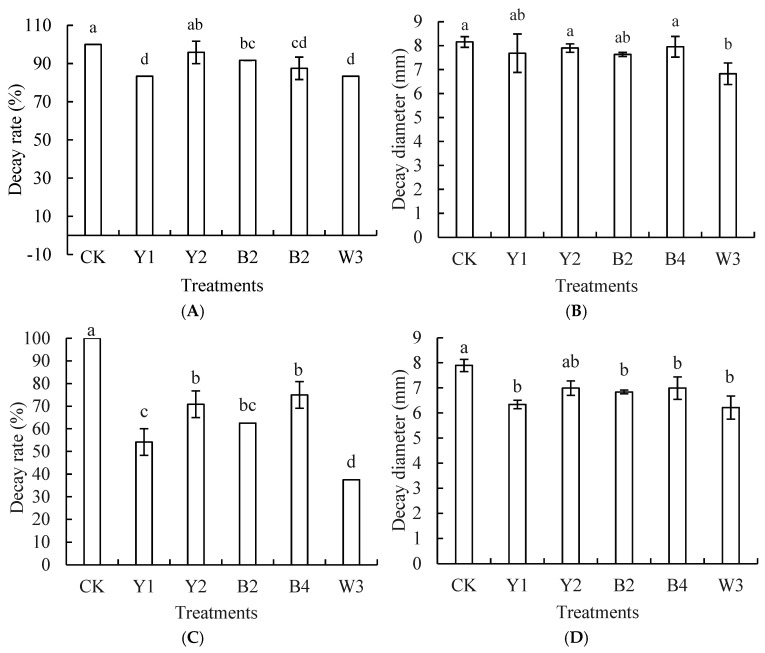
Effects of volatile metabolites produced by different treatments on blue mold (**A**,**B**) and black mold (**C**,**D**) of red grapes. Note: CK is sterile water control group; Y1, Y2, B2, B4, and W3 represent different treatment groups. Different letters represent significant differences (*p* < 0.05) according to Duncan’s multiple-range mean comparison test.

**Figure 8 foods-14-00408-f008:**
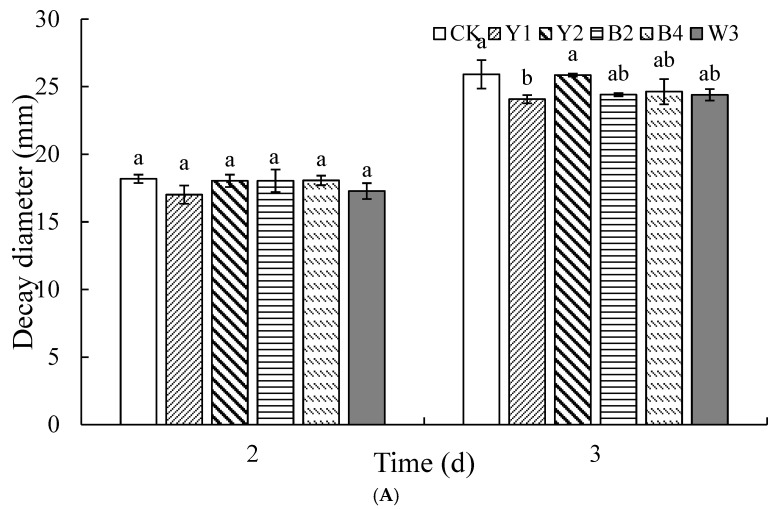
*In vitro* inhibition of *P. expansum* (**A**) and *A. niger* (**B**) by non-volatile metabolites treated with different treatments. Note: CK is sterile water control group; Y1, Y2, B2, B4, and W3 represent different treatment groups. Different letters represent significant differences (*p* < 0.05) according to Duncan’s multiple-range mean comparison test.

**Figure 9 foods-14-00408-f009:**
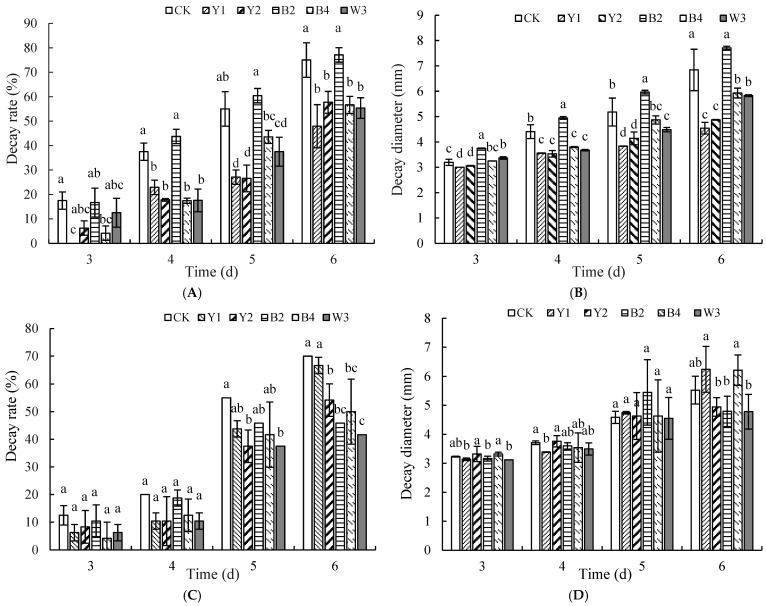
Effects of non-volatile metabolites produced by different treatments on blue mold (**A**,**B**) and black mold (**C**,**D**) of red grapes. Note: CK is sterile water control group; Y1, Y2, B2, B4, and W3 represent different treatment groups. Different letters represent significant differences (*p* < 0.05) according to Duncan’s multiple-range mean comparison test.

**Figure 10 foods-14-00408-f010:**
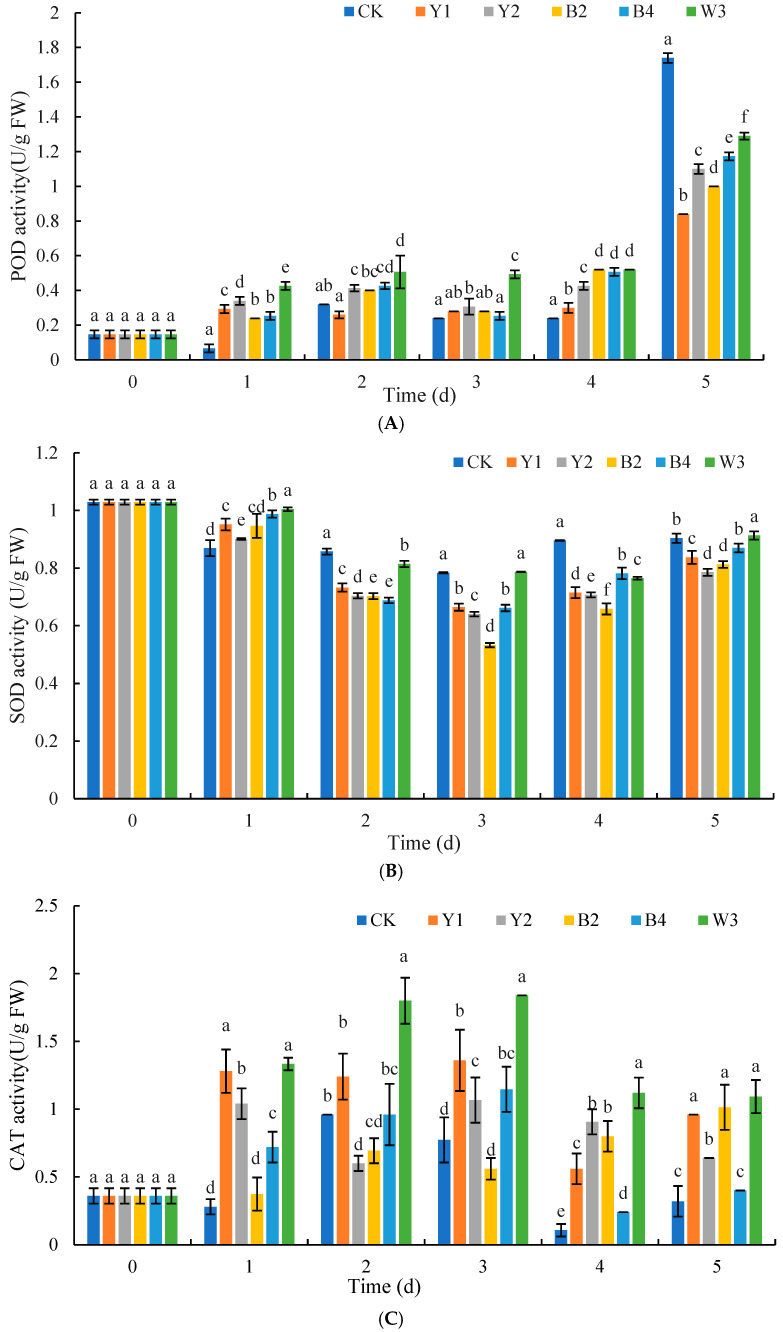
Effects of different treatments on POD (**A**), SOD (**B**) and CAT (**C**) activities of red grapes. Note: CK is sterile water control group; Y1, Y2, B2, B4, and W3 represent different treatment groups. Different letters represent significant differences (*p* < 0.05) according to Duncan’s multiple-range mean comparison test.

**Figure 11 foods-14-00408-f011:**
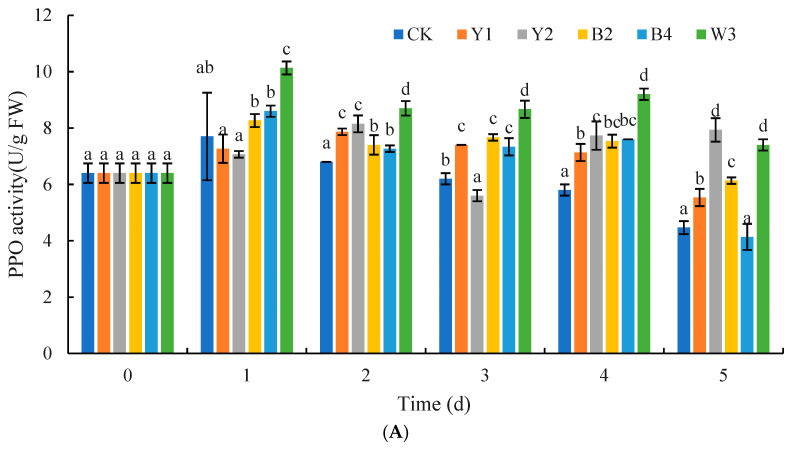
Effects of different treatments on PPO (**A**), APX (**B**), and PAL (**C**) activities of red grapes. Note: CK is sterile water control group; Y1, Y2, B2, B4, and W3 represent different treatment groups. Different letters represent significant differences (*p* < 0.05) according to Duncan’s multiple-range mean comparison test.

**Figure 12 foods-14-00408-f012:**
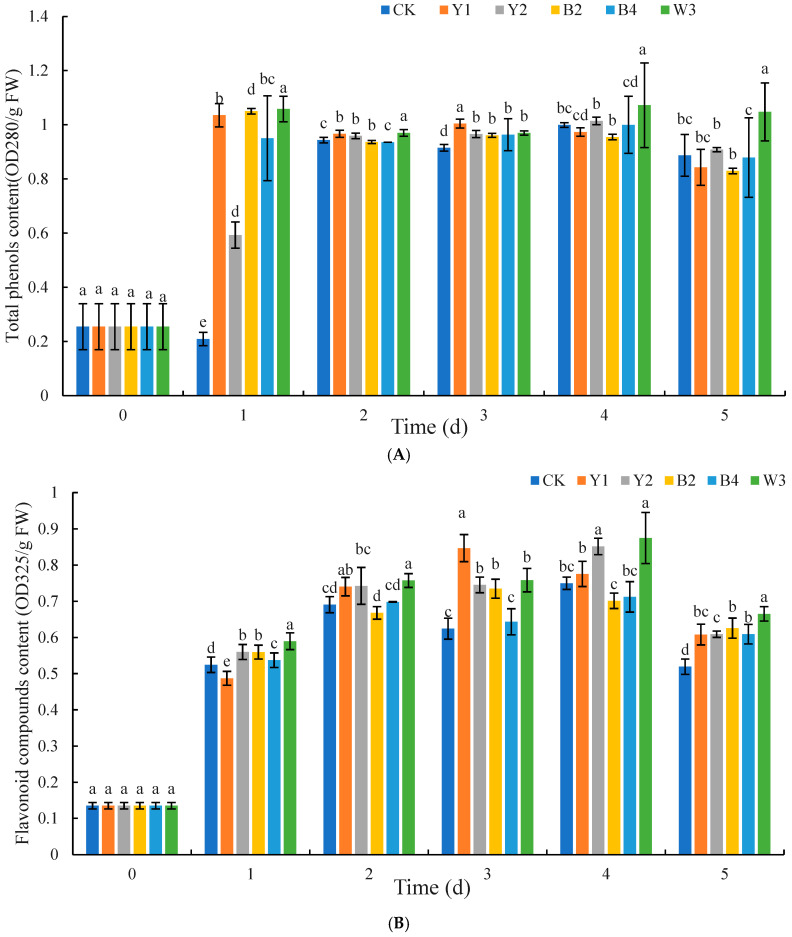
Effects of different treatments on total phenol (**A**) and flavonoid (**B**) contents of red grapes. Note: CK is sterile water control group; Y1, Y2, B2, B4, and W3 represent different treatment groups. Different letters represent significant differences (*p* < 0.05) according to Duncan’s multiple-range mean comparison test.

**Table 1 foods-14-00408-t001:** Construction of complex microorganisms controlling *P. expansum* and *A. niger*.

Pathogenic Fungi	Antagonistic Microorganism	The Ratio of Optimal Biocontrol Effect	Complex Microorganism
*P. expansum*	Y1:B2	1:1	W1
	Y2:W1	1:4	W2
	B4:W2	1:1	W3
*A. niger*	Y1:B2	5:1	Q1
	Y2:Q1	1:1	Q2
	B4:Q2	1:1	Q3

**Table 2 foods-14-00408-t002:** Detection of volatile gases produced by complex microorganism W3.

Volatile Compounds	CAS	Relative Peak Area Ratio (%)	Chemical Formula
Phenylethyl Alcohol	1960/12/8	30.77	C_8_H_10_O
Butanoic acid, 2−methyl−	116−53−0	7.43	C_5_H_10_O_2_
Docosanoic acid, ethyl ester	5908−87−2	4.63	C_24_H_48_O_2_
Butanoic acid, 2−methyl−, ethyl ester	7452−79−1	4.28	C_7_H_14_O_2_
1−Butanol, 3−methyl−, acetate	123−92−2	3.88	C_7_H_14_O_2_
Eicosyl isopropyl ether	0−0−0	3.13	C_6_H_14_O
1−Propanol, 2,2−dimethyl−, acetate	926−41−0	2.86	C_7_H_12_O_4_
Ethyl 13−methyl−tetradecanoate	0−0−0	1.85	C_16_H_32_O_2_
1−Propanol, 2,2−dimethyl−, acetate	926−41−0	2.86	C_7_H_12_O_4_
Nonane, 5−(2−methylpropyl)−	62185−53−9	1.81	C_13_H_28_
2−Methoxy−4−vinylphenol	7786−61−0	1.58	C_9_H_10_O_2_
Phthalan	496−14−0	1.41	C_8_H_8_O
1,4−Butanediamine, 2,3−dimethoxy−N,N,N’,N’−tetramethyl−, [S−(R*,R*)]−	26549−21−3	1.04	C_12_H_23_BN_2_O_4_

Note: R* is for methoxy.

## Data Availability

Data are contained within the article.

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
