# Peer review of "Construction of Composite Microorganisms and Their Physiological Mechanisms of Postharvest Disease Control in Red Grapes"

_foods, 2025, doi:10.3390/foods14030408_

Round 1

Reviewer 1 Report

Comments and Suggestions for Authors

The manuscript entitled “Construction of composite microorganisms and their physiological mechanisms of postharvest disease control in red grapes” explores the use a combination of yeasts and bacteria to control postharvest diseases caused by molds. The research demonstrates that these microbial consortia can effectively reduce spore germination and germ tube growth, while their volatile compounds inhibit pathogen development. Additionally, the combined treatment was shown to enhance the activity and accumulation of antifungal compounds, boosting disease resistance and minimizing decay.

Although the study offers valuable insights into natural biocontrol strategies, certain aspects require substantial revision to enhance the clarity, strengthen scientific rigor, and maximize the manuscript’s overall impact. A language revision is also essential. Therefore, I recommend major revision and resubmission of the work.

Below are the detailed comments:

It is recommended to carefully review the manuscript and address some formatting issues to ensure adherence to standard scientific writing conventions (e.g. lines 13, 297, 399, 445, figure 6, 7, 8, 9, 12).

Introduction

In the introduction, I recommend providing more detailed information on Rhodotorula and Bacillus strains, with a focus on the antimicrobial compounds they produce. Elaborate on the types of antimicrobial substances these microorganisms can synthesize and their mechanisms of action. Additionally, clarify the rationale behind selecting these specific yeast and bacterial strains for study (for example, these microorganisms may be naturally associated with grapes or commonly found in vineyards, contributing to their selection for study). This will enhance the reader’s understanding of their potential efficacy and relevance in biocontrol applications.

Methods

I noticed that the methods section does not include details about the statistical analyses used in the study. Could you please add the missing information regarding the statistical methods employed?

Lines 93-96

I am curious about how the species of isolated molds were identified. Could you clarify what is meant by the term mycobacteria (line 95) in the context of spore suspension preparation? Additionally, the section on pathogens lacks sufficient detail and would benefit from further elaboration.

Lines 97-108

Is the data in lines 98-99 derived from another study? If so, please provide the appropriate reference. The text lacks information on how it was determined that these species exhibit biocidal properties.

Lines 120-153

I noticed that the protocols described in sections 2.4.2. and 2.4.3. appear to be identical, aside from mold species involved. Could you consider merging these two sections (2.4.2 and 2.4.3) into a single chapter to avoid redundancy? Additionally, in lines 152-153 you mention that “the antagonistic microorganism and red grapes were treated as described in Section 1.7.1. However, I could not locate a by that designation. Please correct this reference.

Line 193

You stated, “In detail, 5 mL of PDA medium was added in a sterile sealed gas bottle, and after it solidified, 20 μL of bacterial suspensions of Y1, Y2, B2, B4, and W3”. However, this is not accurately described as a bacterial suspension, as it consists of a mixture of both yeasts and bacteria. Please revise this to reflect that it is a microorganism suspension or composite.

Lines 222-223

I am a bit confused. You mentioned, “The treatment of red grapes followed the same procedure as in Section 2.6, and the 222 preparation of the antagonist supernatant was also as described in Section 2.6.3.”. However, I believe this may be an error. It seems that the preparation of the antagonist supernatant aligns more closely with the procedure described in Section 2.6.2. Please review and correct this if necessary.

Lines 237-238

I recommend revising the sentence for clarity and grammatical accuracy. The current phrasing,” Then, the red grapes from the wound was removed, and 0.5g of fruit sample was scooped from the outer rim of the wound, quickly frozen with liquid nitrogen, and then placed at -80°C for backup” appears unclear and contains grammatical errors. A clearer and more precise revision could be: “the affected area of the red grapes around the wound was removed”.

Lines 281-282

The sentence “Fruit pulp (0.5 g) was mixed with 1 mL of 1% HCl-methanol solution was extracted in an ice bath and protected from light for 20 min” is grammatically incorrect, please rewrite it, for example “Fruit pulp (0.5 g) was mixed with 1 mL of 1% HCl-methanol solution, extracted in an ice bath, and protected from light for 20 min.”

Results

The construction of the optimal ratio of composite microorganisms is highly interesting. However, I recommend including a table detailing the resulting composites to clarify which microorganisms and their respective ratios correspond to W1, W2, W3, Q1, Q2 and Q3. This will enhance the clarity and understanding of the experimental design.

Figure 2 and 3

I recommend addressing some formatting issues in these figures. Additionally, I have a question regarding the boxes at the top of certain columns – could you clarify their purpose? Similar structures do not appear in the other figures.

Lines 376-382

I found lines 376-378 somewhat difficult to follow. The text would be clearer if the comparisons with the control were made directly, rather than discussed separately. I suggest presenting the percentage alongside the control percentage and comparing germ tube lengths directly with the germ tube length of the control.

Lines 377 and Figure 5 (B and D)

In line 377, you refer to “germ tube length,” while Figure 5 (B and D) uses the term “bud tube”. I believe these refer to the same structure. Could you please standardize the terminology for consistency throughout the text?

Line 399 and Figure 6

You mentioned that “Figure 6C and 6D demonstrate the curbing influence on A. niger”. However, I could not locate panels 6C and 6D within Figure 6. Additionally, the content of the image is unclear due to the absence of a corresponding description. Furthermore, the labels next to the image read “B1” and “B2”, whereas the bacterial strains referenced earlier were identified as “B2” and “B4”. And finally, the term “curbing influence” could be replaced by more precise and scientific terms, for example: inhibitory effect, suppressive effect, growth inhibition and cetera. Could you clarify if these refer to the same strains and correct any inconsistencies?

Lines 432-439

I recommend avoiding repetition of the content already presented in Table1. Consider either listing all volatile compounds in the table or describing them in the text, but not both. This will improve clarity and reduce redundancy.

Line 445

You mentioned that “Figure 3 shows the in vitro inhibitory outcomes of non-volatile metabolites”. I believe this is a mistake, and you likely meant Figure 8. Additionally, “in vitro” should be written in italics. Please review and correct this accordingly.

Figure 9

I recommend reformatting this image. I believe that 9C should represent decay rate (%) rather than decay diameter (mm). Currently, the diagrams in 6B, 6C and 6D display decay diameter, while only 6A shows decay rate.

Discussion

I believe your results warrant a more detailed discussion. You have presented a substantial amount of data, but I noticed a lack of focus on comparing your findings with those of other researchers, for example with those who have also used composite microorganisms. Currently, you mention that previous studies have shown that the compound microorganisms showed better efficacy in controlling molds, but this point could benefit from deeper analysis and discussion.  

Additionally, it remains unclear how you selected these specific strains for your research and what evidence supports their superior biocontrol efficacy. The text lacks citations or references to your previous work, which would help justify these choices.  

I also found limited information regarding the non-volatile compounds produced by the bacteria and yeast. There is no mention of specific non-volatile substances that could contribute to antifungal activity. Please consider expanding this part of discussion by identifying potential secreted compounds and explaining why you did not investigate whether your strains produce toxins or bacteriocins.

What types of non-volatile substances could play a role in the biocontrol of A. niger and P. expansum?

Please consider expanding your discussion, as your research is both highly interesting and valuable.

Comments on the Quality of English Language

The English must be improved to more clearly express the research.

Author Response

Dear reviewer:

We would like to thank you for your precious time and effort in providing valuable comments on our manuscript. The feedback and suggestions have helped us improve the quality and content of our manuscripts. The changes made in the revised manuscript are highlighted in red. Here, we respond to the reviewer's comments point by point.

Comments: Although the study offers valuable insights into natural biocontrol strategies, certain aspects require substantial revision to enhance the clarity, strengthen scientific rigor, and maximize the manuscript’s overall impact. A language revision is also essential. Therefore, I recommend major revision and resubmission of the work.

  1. It is recommended to carefully review the manuscript and address some formatting issues to ensure adherence to standard scientific writing conventions (e.g. lines 13, 297, 399, 445, figure 6, 7, 8, 9, 12).

Response: Thank you dear reviewer for your helpful suggestion. We carefully revised the formatting issues mentioned and others in our revised manuscript. Major changes made are marked in red color.

  1. Introduction

In the introduction, I recommend providing more detailed information on Rhodotorula and Bacillus strains, with a focus on the antimicrobial compounds they produce. Elaborate on the types of antimicrobial substances these microorganisms can synthesize and their mechanisms of action. Additionally, clarify the rationale behind selecting these specific yeast and bacterial strains for study (for example, these microorganisms may be naturally associated with grapes or commonly found in vineyards, contributing to their selection for study). This will enhance the reader’s understanding of their potential efficacy and relevance in biocontrol applications.

Response: Thank you dear reviewer for your important recommendation.

Based on your comments, we have added the description of Rhodotorula sp. and Bacillus sp. and the antibacterial substances produced by them in the introduction, and explained the reasons for selecting them in line 45-63.

  1. I noticed that the methods section does not include details about the statistical analyses used in the study. Could you please add the missing information regarding the statistical methods employed?

Response: Thank you dear reviewer for your helpful suggestion. We have added the statistical method employed and mean comparison methods used in our revised manuscript line 294-296.

  1. Lines 93-96

I am curious about how the species of isolated molds were identified. Could you clarify what is meant by the term mycobacteria (line 95) in the context of spore suspension preparation? Additionally, the section on pathogens lacks sufficient detail and would benefit from further elaboration.

Response: Thank you dear reviewer for your helpful suggestion.

With regard to term the term mycobacteria (line 95), we made a mistake while writing the manuscript and it has now been corrected. Both P. expansum and A. niger were isolated from the surface of rotten grapes with our research team and preserved in the laboratory. We used those strains for the current study. We revised this information in our revised manuscript lines 108-111.

5.Lines 97-108

Is the data in lines 98-99 derived from another study? If so, please provide the appropriate reference. The text lacks information on how it was determined that these species exhibit biocidal properties.

Response: Thank you dear reviewer for your important recommendation.

Based on your comments, we have added information about these four strains to the manuscript in line 116-119.

As to whether this is another study, I would like to answer yes, but the content of this part has been put in another manuscript which is not yet published. In our study, we first analyzed the microbial communities on the surface of red grapes with different storage times under natural conditions, and combined with the analysis results, screened some species that could not only inhibit pathogenic fungi but also have high abundance on the surface of red grapes. Based on this, we were able to screen these four strains and rank their biocontrol effectiveness. However, since the biocontrol efficacy of a single strain has been included in the previous study manuscript which is currently under review, we cannot cite or use the data in this work.

  1. Lines 120-153

I noticed that the protocols described in sections 2.4.2. and 2.4.3. appear to be identical, aside from mold species involved. Could you consider merging these two sections (2.4.2 and 2.4.3) into a single chapter to avoid redundancy? Additionally, in lines 152-153 you mention that “the antagonistic microorganism and red grapes were treated as described in Section 1.7.1. However, I could not locate a by that designation. Please correct this reference.

Dear reviewer, thanks for your helpful suggestion. As suggested, we have merged this part of the content and modified the references in lines 148-156.

  1. Line 193

You stated, “In detail, 5 mL of PDA medium was added in a sterile sealed gas bottle, and after it solidified, 20 μL of bacterial suspensions of Y1, Y2, B2, B4, and W3”. However, this is not accurately described as a bacterial suspension, as it consists of a mixture of both yeasts and bacteria. Please revise this to reflect that it is a microorganism suspension or composite.

Dear reviewer, thanks for your helpful suggestion. We have revised the description of this part of the content in line 197-198.

  1. Lines 222-223

I am a bit confused. You mentioned, “The treatment of red grapes followed the same procedure as in Section 2.6, and the 222 preparation of the antagonist supernatant was also as described in Section 2.6.3.”. However, I believe this may be an error. It seems that the preparation of the antagonist supernatant aligns more closely with the procedure described in Section 2.6.2. Please review and correct this if necessary.

Dear reviewer, thanks for your careful checks. We have checked and modified the reference in this part in line 227 and 228.

  1. Lines 237-238

I recommend revising the sentence for clarity and grammatical accuracy. The current phrasing,” Then, the red grapes from the wound was removed, and 0.5g of fruit sample was scooped from the outer rim of the wound, quickly frozen with liquid nitrogen, and then placed at -80°C for backup” appears unclear and contains grammatical errors. A clearer and more precise revision could be: “the affected area of the red grapes around the wound was removed”.

Dear reviewer, thank you for your recommendation. We have revised the description of this sentence in line 242-245.

  1. Lines 281-282

The sentence “Fruit pulp (0.5 g) was mixed with 1 mL of 1% HCl-methanol solution was extracted in an ice bath and protected from light for 20 min” is grammatically incorrect, please rewrite it, for example “Fruit pulp (0.5 g) was mixed with 1 mL of 1% HCl-methanol solution, extracted in an ice bath, and protected from light for 20 min.”

Dear reviewer, thank you for your recommendation. We have revised the description of this sentence in line 286-287.

  1. Results

The construction of the optimal ratio of composite microorganisms is highly interesting. However, I recommend including a table detailing the resulting composites to clarify which microorganisms and their respective ratios correspond to W1, W2, W3, Q1, Q2 and Q3. This will enhance the clarity and understanding of the experimental design.

Thank you, dear reviewer, for your helpful suggestion.

Based on the comments, we have added this part of the presentation in tabular form (Table 1 line 375). It is hoped to enhance the clarity of the experimental design.

  1. Figure 2 and 3

I recommend addressing some formatting issues in these figures. Additionally, I have a question regarding the boxes at the top of certain columns – could you clarify their purpose? Similar structures do not appear in the other figures.

Dear reviewer, thanks for your helpful suggestion. We have checked and corrected the picture problems in the manuscript.

  1. Lines 376-382

I found lines 376-378 somewhat difficult to follow. The text would be clearer if the comparisons with the control were made directly, rather than discussed separately. I suggest presenting the percentage alongside the control percentage and comparing germ tube lengths directly with the germ tube length of the control.

Dear reviewer, thank you for your important recommendation. Based on the comments, we have revised the description of this sentence in lines 389-392.

  1. Lines 377 and Figure 5 (B and D)

In line 377, you refer to “germ tube length,” while Figure 5 (B and D) uses the term “bud tube”. I believe these refer to the same structure. Could you please standardize the terminology for consistency throughout the text?

Thank you, dear reviewer, for your helpful suggestion. We have changed the word "bud tube " in the manuscript to " germ tube length ".

  1. Line 399 and Figure 6

You mentioned that “Figure 6C and 6D demonstrate the curbing influence on A. niger”. However, I could not locate panels 6C and 6D within Figure 6. Additionally, the content of the image is unclear due to the absence of a corresponding description. Furthermore, the labels next to the image read “B1” and “B2”, whereas the bacterial strains referenced earlier were identified as “B2” and “B4”. And finally, the term “curbing influence” could be replaced by more precise and scientific terms, for example: inhibitory effect, suppressive effect, growth inhibition and cetera. Could you clarify if these refer to the same strains and correct any inconsistencies?

Dear reviewer, thank you for pointing out our mistake in this figure. In our revised manuscript, we corrected these problems in Fig. 6.

  1. Lines 432-439

I recommend avoiding repetition of the content already presented in Table1. Consider either listing all volatile compounds in the table or describing them in the text, but not both. This will improve clarity and reduce redundancy.

Dear reviewer, thank you for your helpful suggestion. We have cut this part of the text and only kept the first three presentations.

  1. Line 445

You mentioned that “Figure 3 shows the in vitro inhibitory outcomes of non-volatile metabolites”. I believe this is a mistake, and you likely meant Figure 8. Additionally, “in vitro” should be written in italics. Please review and correct this accordingly.

Response: Thank you dear reviewer for your helpful suggestion.

Dear reviewer, thank you for pointing out our mistake. We checked and corrected these errors in the manuscript.

  1. Figure 9

I recommend reformatting this image. I believe that 9C should represent decay rate (%) rather than decay diameter (mm). Currently, the diagrams in 6B, 6C and 6D display decay diameter, while only 6A shows decay rate.

Dear reviewer, thank you for pointing out our mistake. We have corrected the figures in our revised manuscript.

  1. Discussion

I believe your results warrant a more detailed discussion. You have presented a substantial amount of data, but I noticed a lack of focus on comparing your findings with those of other researchers, for example with those who have also used composite microorganisms. Currently, you mention that previous studies have shown that the compound microorganisms showed better efficacy in controlling molds, but this point could benefit from deeper analysis and discussion. 

Dear reviewer, I suggested we improved the discussion section of our revised manuscript. Major changes made are marked in red color.

  1. Additionally, it remains unclear how you selected these specific strains for your research and what evidence supports their superior biocontrol efficacy. The text lacks citations or references to your previous work, which would help justify these choices.

Dear reviewer, thank you for the important question. In our study, we first analyzed the microbial communities on the surface of red grapes with different storage times under natural conditions, and combined with the analysis results, screened some species that could not only inhibit pathogenic fungi but also have high abundance on the surface of red grapes. Based on this, we were able to screen these four strains and rank their biocontrol effectiveness. However, since the biocontrol efficacy of single strains has been already determined by other researcher from our lab and the data is not yet published we couldn’t cite the work in this study. We have also conducted some preliminary works to confirm the result of the individual efficiency.

  1. I also found limited information regarding the non-volatile compounds produced by the bacteria and yeast. There is no mention of specific non-volatile substances that could contribute to antifungal activity. Please consider expanding this part of discussion by identifying potential secreted compounds and explaining why you did not investigate whether your strains produce toxins or bacteriocins.

What types of non-volatile substances could play a role in the biocontrol of A. niger and P. expansum?

Dear reviewer, thank you for the important question. Because in our study, we did not find that the non-volatile substances of complex microorganisms had obvious inhibitory effect on pathogenic fungi, compared with all the single antagonistic strains. Therefore, we have not studied its non-volatile substances, and the research on this part is expected to be carried out in the future.

  1. Please consider expanding your discussion, as your research is both highly interesting and valuable.

Dear reviewer, thank you very much for your recognition of our research content and your careful review of the manuscript content. We have revised and added the content and improved the grammar of the discussion and other sections, which the major changes are marked in red color in our revised manuscript.

Reviewer 2 Report

Comments and Suggestions for Authors

Overall, the work is well-structured, and this contribution should consider the following comments: 

1. The introduction covers the use of microbial agents in controlling two important diseases in grapes. However, the authors state that the use of pesticides causes resistance in phytopathogens and causes problems for human health. The use of microorganisms is always talked about only for disease control and reduction of pesticides. The question is whether these microbial agents are completely safe for human health? There are mycotoxins produced by various microorganisms that are toxic to humans. Therefore, it is important to be cautious regarding these statements.

2. Are the isolates of P. expansum and A. niger used monosporic cultures? Was a molecular identification performed? Are they deposited in GenBank? Please include this data, which is important to validate the research.

3. Discussion - The use of microorganisms in post-harvest disease control is not a prevalent alternative, but rather an alternative that is being tested.

4. If there was no substantial distinction between treatment with a consortium of microorganisms and treatment with individual antagonists, why so much emphasis on the mixture of microorganisms in the text?

5. The discussions are largely based on results from other authors, with other pathosystems and other microbial biocontrol agents. I recommend that the authors give more emphasis to their results. The discussion deserves to be improved, the work is good and innovative, but the discussions do not reflect the importance of the work.

Author Response

Dear reviewer:

We would like to thank you for your precious time and effort in providing valuable comments on our manuscript. The feedback and suggestions have helped us improve the quality and content of our manuscripts. The changes made in the revised manuscript are highlighted in red. Here, we respond to the reviewer's comments point by point.

Comments:

  1. The introduction covers the use of microbial agents in controlling two important diseases in grapes. However, the authors state that the use of pesticides causes resistance in phytopathogens and causes problems for human health. The use of microorganisms is always talked about only for disease control and reduction of pesticides. The question is whether these microbial agents are completely safe for human health? There are mycotoxins produced by various microorganisms that are toxic to humans. Therefore, it is important to be cautious regarding these statements.

Dear reviewer, thank you for your excellent suggestion. As stated, some microorganisms are dangerous to human health, even worse than the chemical pesticides by producing toxic compounds such as mycotoxins. When selecting the antagonistic microbes, their safety should be a priority. Here in our lab, we isolated few microorganisms (yeasts and bacteria mostly) which have antagonistic characteristics and can inhibit the growth of other pathogenic microbes. We conducted the non-toxicity test for all of them including the strains used in the current study. We have included information regarding the safety of biocontrol to emphasize this issue in our revised manuscript lines 42-44.

  1. Are the isolates of P. expansumand A. nigerused monosporic cultures? Was a molecular identification performed? Are they deposited in GenBank? Please include this data, which is important to validate the research.

Response: Thank you so much for your helpful suggestion. Dear reviewer, thank you for pointing out excellent question. Yes, both fungi were isolated from the surface of rotten red grapes, cultured as single-spore and finally the strains were identified by molecular techniques. We preserved the strains in our laboratory under -80 °C which were then activated and used for this experiment. We have included this information in our revised manuscript line 107-111.  

  1. Discussion - The use of microorganisms in post-harvest disease control is not a prevalent alternative, but rather an alternative that is being tested.

Dear reviewer, thank you for the excellent suggestion. We corrected the description of the statement in our revised manuscript lines 587-589.

  1. If there was no substantial distinction between treatment with a consortium of microorganisms and treatment with individual antagonists, why so much emphasis on the mixture of microorganisms in the text?

Dear reviewer, thank you for pointing out an important question. Our results showed that the control effect of composite microorganisms on both P. expansum and A. niger was significantly higher than that of single antagonistic microorganisms. In terms of controlling spore germination of P. expansum and A. niger, inhibiting pathogenic fungi by producing volatile substances, and inducing resistance of red grape fruit itself, the composite microorganisms showed significantly better biocontrol efficacy than single antagonistic microorganisms. In addition to the inhibition of pathogenic fungi by the non-volatile substances produced by them, some single antagonistic microorganisms showed similar biocontrol efficacy as the composite microorganisms, so we can also infer that the reason why the composite microorganisms showed better biocontrol efficacy than the single antagonistic microorganisms. Because complex microbes can do better in other ways than producing non-volatile substances.

  1. The discussions are largely based on results from other authors, with other pathosystems and other microbial biocontrol agents. I recommend that the authors give more emphasis to their results. The discussion deserves to be improved, the work is good and innovative, but the discussions do not reflect the importance of the work.

Dear reviewer, thank you for your appreciation of our research content. As you suggested, we have supplemented and revised the content of the discussion section, which is marked in red color in our revised manuscript.

Round 2

Reviewer 1 Report

Comments and Suggestions for Authors

While the majority of the comments have been addressed, several errors remain that I would like the authors of the article to rectify. This primarily pertains to newly introduced errors in the text and numerous formatting issues. Therefore, I recommend minor revision and resubmission of the work.

Below are the detailed comments:

It is recommended to carefully review the manuscript and address some formatting issues to ensure adherence to standard scientific writing conventions (for example, in Figure 3, the notes by the graphs are misaligned, one graph is positioned lower than the other, and some letters overlap with the numbers. Similar issues are observed in Figures 4, 5, 6, 7, 8, 9, 10, 11).

Lines 141, 167, 181, 189, 191 formatting issues.

Lines 150-151

You wrote: “Then the optimal composite proportion between the next antagonist antagonistic microorganisms and.” This phrasing is grammatically incorrect. A corrected version would be “. This phrasing is grammatically incorrect. A corrected version would be: Then the optimal composite proportions between the subsequent antagonistic microorganisms and .”

Lines 242-244

Sentence: “Then, the affected area of red grapes around the wound was removed, the red grapes from the wound was removed, and 0.5g of fruit sample was extracted from the outer edge of the wound, quickly frozen with liquid nitrogen, and then placed at -80°C for backup” contains some redundancies and minor grammatical issues. A clearer and more precise revision could be: “The affected area of red grapes around the wound was removed, the tissue from the wound was excised, and 0.5g of fruit sample was extracted from the outer edge of the wound, quickly frozen with liquid nitrogen, and then placed at -80°C for future use”

Lines 371-373

The sentence “Therefore, the combined effects of two composite microorganisms, W3 and Q3, on P. expansum and A. niger control effect, we finally chose W3 as the optimal formulation and used it for subsequent experiments” are a little bit unclear. The corrected version would be: “Therefore, based on the combined effects of the two composite microorganisms, W3 and Q3, on the control of P. expansum and A. niger, we selected W3 as the optimal formulation for subsequent experiments

Line 375 Table 1.

I believe that there is an error in this table. In the second row, under the section for antagonistic microorganisms and A. niger, it should list Y1:B2, Y2:Q1 and B4:Q2, rather than repeating the first set of data.

Lines 391 and 395 and Figure 5 (B and D)

In line 391 and 395, you refer to “germ tube length,” while Figure 5 (B and D) uses the term “bud tube length”. I believe these refer to the same structure. Could you please standardize the terminology for consistency throughout the text and graph?

Line 419 and Figure 6

The notes above and next to the graph are too small, please increase the font size.

Comments on the Quality of English Language

The English could be improved to more clearly express the research.

Author Response

Dear reviewer:

We would like to thank you for your precious time and effort in providing valuable comments on our manuscript again. The feedback and suggestions have helped us improve the quality and content of our manuscripts. The changes made in the revised manuscript are highlighted in green. Here, we respond to your comments point by point.

Comments:

  1. It is recommended to carefully review the manuscript and address some formatting issues to ensure adherence to standard scientific writing conventions (for example, in Figure 3, the notes by the graphs are misaligned, one graph is positioned lower than the other, and some letters overlap with the numbers. Similar issues are observed in Figures 4, 5, 6, 7, 8, 9, 10, 11).

Thank you, dear reviewer, for your helpful suggestion. We carefully revised the formatting issues mentioned and others in our revised manuscript. As for the diagram section, we have also revised it and presented it in the manuscript Major changes made are marked in green color.

  1. Lines 141, 167, 181, 189, 191 formatting issues.

Dear reviewer, thanks for your comment. We have checked and modified the formatting issues.

  1. Lines 150-151

You wrote: “Then the optimal composite proportion between the next antagonist antagonistic microorganisms and.” This phrasing is grammatically incorrect. A corrected version would be “. This phrasing is grammatically incorrect. A corrected version would be: Then the optimal composite proportions between the subsequent antagonistic microorganisms and .”

Dear reviewer, thank you for your recommendation. We have revised the description of this sentence in line 150-151 of our revised manuscript.

  1. Lines 242-244

Sentence: “Then, the affected area of red grapes around the wound was removed, the red grapes from the wound was removed, and 0.5g of fruit sample was extracted from the outer edge of the wound, quickly frozen with liquid nitrogen, and then placed at -80°C for backup” contains some redundancies and minor grammatical issues. A clearer and more precise revision could be: “The affected area of red grapes around the wound was removed, the tissue from the wound was excised, and 0.5g of fruit sample was extracted from the outer edge of the wound, quickly frozen with liquid nitrogen, and then placed at -80°C for future use”

Dear reviewer, thank you for your recommendation. We have revised the description of this sentence in line 243-246 of our revised manuscript.

  1. Lines 371-373

The sentence “Therefore, the combined effects of two composite microorganisms, W3 and Q3, on P. expansum and A. niger control effect, we finally chose W3 as the optimal formulation and used it for subsequent experiments” are a little bit unclear. The corrected version would be: “Therefore, based on the combined effects of the two composite microorganisms, W3 and Q3, on the control of P. expansum and A. niger, we selected W3 as the optimal formulation for subsequent experiments

Dear reviewer, thank you for your recommendation. We have revised the description of this sentence in line 385-387 of our revised manuscript.

  1. Line 375 Table 1.

I believe that there is an error in this table. In the second row, under the section for antagonistic microorganisms and A. niger, it should list Y1:B2, Y2:Q1 and B4:Q2, rather than repeating the first set of data.

Dear reviewer, thank you for your helpful suggestion. We made a mistake while writing the manuscript and it has now been corrected. We revised this information in our revised manuscript lines 388-389.

  1. Lines 391 and 395 and Figure 5 (B and D)

In line 391 and 395, you refer to “germ tube length,” while Figure 5 (B and D) uses the term “bud tube length”. I believe these refer to the same structure. Could you please standardize the terminology for consistency throughout the text and graph?

Thank you, dear reviewer, for your helpful suggestion. We have changed the word "bud tube " in the manuscript to " germ tube length ".

  1. Line 419 and Figure 6

The notes above and next to the graph are too small, please increase the font size.

Thank you, dear reviewer, for your helpful suggestion. We have revised the figures according to the comment and presented it in line 438 of our revised manuscript.